# Feasibility of Portable Microwave Imaging Device for Breast Cancer Detection

**DOI:** 10.3390/diagnostics12010027

**Published:** 2021-12-23

**Authors:** Mio Adachi, Tsuyoshi Nakagawa, Tomoyuki Fujioka, Mio Mori, Kazunori Kubota, Goshi Oda, Takamaro Kikkawa

**Affiliations:** 1Department of Surgery, Breast Surgery, Tokyo Medical and Dental University, Tokyo 113-8510, Japan; mioadachi1016@gmail.com (M.A.); odasrg2@tmd.ac.jp (G.O.); 2Department of Diagnostic Radiology, Tokyo Medical and Dental University, Tokyo 113-8510, Japan; fjokmrad@tmd.ac.jp (T.F.); m_mori_116@yahoo.co.jp (M.M.); kbtmrad@tmd.ac.jp (K.K.); 3Department of Radiology, Dokkyo Medical University, Tochigi 321-0293, Japan; 4Research Institute for Nanodevice and Bio Systems, Hiroshima University, Hiroshima 739-8527, Japan; kikkawat@hiroshima-u.ac.jp

**Keywords:** breast cancer, microwave imaging, detectability, screening, ultrawideband radar, dome antenna

## Abstract

Purpose: Microwave radar-based breast imaging technology utilizes the principle of radar, in which radio waves reflect at the interface between target and normal tissues, which have different permittivities. This study aims to investigate the feasibility and safety of a portable microwave breast imaging device in clinical practice. Materials and methods: We retrospectively collected the imaging data of ten breast cancers in nine women (median age: 66.0 years; range: 37–78 years) who had undergone microwave imaging examination before surgery. All were Japanese and the tumor sizes were from 4 to 10 cm. Using a five-point scale (1 = very poor; 2 = poor; 3 = fair; 4 = good; and 5 = excellent), a radiologist specialized in breast imaging evaluated the ability of microwave imaging to detect breast cancer and delineate its location and size in comparison with conventional mammography and the pathological findings. Results: Microwave imaging detected 10/10 pathologically proven breast cancers, including non-invasive ductal carcinoma in situ (DCIS) and micro-invasive carcinoma, whereas mammography failed to detect 2/10 breast cancers due to dense breast tissue. In the five-point evaluation, median score of location and size were 4.5 and 4.0, respectively. Conclusion: The results of the evaluation suggest that the microwave imaging device is a safe examination that can be used repeatedly and has the potential to be useful in detecting breast cancer.

## 1. Introduction

The number of breast cancer cases in Japan has been increasing since the 2000s and is currently the most common cancer in Japanese women [1]. The number of breast cancers and deaths in women has increased annually in Japan [1].

Mammography (MG) is commonly used for breast cancer screening, which has been shown to decrease the mortality rate of breast cancer [2]. However, MG has several problems. Dense breast tissue makes it difficult to detect breast cancer on MG and reduces its sensitivity [3]. Young women and Asian women commonly have dense breast tissue and may not be suitable for MG screening [4]. In addition, MG is a painful examination, with radiation exposure [5].

It is known that adding ultrasound (US) to MG increases sensitivity; however, US has the problems of a high false-positive rate and that the diagnostic accuracy depends on the skill of the operator [6]. Dynamic contrast-enhanced (DCE)-magnetic resonance imaging (MRI) has very high sensitivity but low specificity [7]. In addition, repeated use of gadolinium contrast media causes side effects, increases costs, and deposits gadolinium in the body [8]. DCE-MRI screening is useful for high-risk breast cancer patients, but is considered over-testing in non-high-risk patients [8]. Although 18F-fluorodeoxyglucose (FDG)/positron emission tomography (PET)-computed tomography (CT) is excellent for staging advanced breast cancer, it is not suitable for screening due to its limited detection of tumors with low metabolic activity and small breast cancers [9,10].

There is a need for a device that could replace MG and improve the breast cancer screening experience, without pain or radiation exposure. A device with the ability to detect breast cancer with performance independent of breast composition is particularly desirable.

A recent study has reported that the electrical and dielectric properties of breast cancer differ from those of normal tissue [11]. Accordingly, a microwave radar-based imaging technique has been developed for breast imaging. This technique can detect breast cancer by measuring the time of flight of reflected microwaves. Microwave imaging is free of ionizing radiation and has been proposed as an alternative to MG screening; however, the conventional prototypes employ vector network analyzers that require heavy instruments and have high costs [12,13]. We have developed a compact and light device for breast tumor detection that employs complementary metal-oxide-semiconductor (CMOS)-integrated circuits and a microwave radar-based imaging system [14,15,16,17,18]. This imaging method is free of ionizing radiation and we expect that it could be used as an alternative to screening MG.

The aim of this study was to assess the ability of a newly developed device that uses non-invasive radar-based microwave technology to detect breast cancer in real patients by using it for more patients than before in more cases, including cancers of various sizes and pathologies, and whether they can be used in a different facility.

## 2. Materials and Methods

### 2.1. Patients

The Medical Ethics Committee of our hospital approved this retrospective study and we obtained written informed consent from all patients.

This study included women aged > 20 years who were diagnosed with breast cancer and scheduled for mastectomy at our hospital between September 2017 and March 2018. Patients who had undergone breast tumor resection on the ipsilateral side in the past, who were pregnant or could possibly be pregnant, or were breast feeding were excluded from the study.

### 2.2. MG, US, DCE-MRI, and FDG-PET/CT Protocol

Prior to surgery, all patients underwent MG and US, and some additionally underwent DCE-MRI and FDG-PET/CT imaging examinations, according to the discretion of the breast surgeon.

MG (craniocaudal and mediolateral oblique views) was performed using Amulet Innovality (FujiFilm, Tokyo, Japan), and US examinations were conducted using an Aplio 500 US machine with a PLT-805AT 8.0-MHz linear probe (Toshiba Medical Systems, Tochigi Prefecture, Japan). MRI of both breasts was acquired using a 3.0-T Signa HDxt system (General Electric Medical Systems, Milwaukee, Brookfield, WI, USA) with a breast coil and the patient in the prone position. Unenhanced and enhanced phases were acquired at 1, 2, and 6 min in the axial plane after intravenous bolus injection of gadobutrol (0.1 mL/kg), using a fat-suppressed T1-weighted sequence (TR/TE = 6.5/2.4, flip angle = 10°, 2 mm thick section, 512 × 512 matrix, 360 mm field of view). Whole-body PET imaging was obtained 60 min after intravenous administration of 18F-FDG (3.7 MBq/kg; 0.1 mCi/kg) using a Celesteion PET/CT system (Canon Medical Systems, Tokyo, Japan).

### 2.3. Microwave Imaging Protocol

Microwave radar-based breast imaging technology utilizes the principle of radar, in which radio waves reflect at the interface between target and normal tissues that have different permittivities [11]. By calculating the time of flight between the transmitting and receiving antennas, we can infer a target on an elliptical trajectory focusing on these two antennas. The position of the target can be estimated by computing the intersection of the trajectories of multiple antennas [13] (Figure 1).

The hand-held microwave imaging device evaluated here has been described in detail previously [14,15,16,17,18]. It includes a handle, stepping motor, control module, radio-frequency module, and dome antenna within a 191 × 177 × 188 mm device and weighs 2 kg. The core functional part of the detector comprises 65-nm technology CMOS integrated circuits covering the ultrawideband width of 3.1–10.6  GHz, which enables the generation and transmission of Gaussian monocycle pulse (GMP) and single port eight throw switching matrices for controlling the 4 × 4 cross-shaped dome antenna array (Figure 2 and Figure 3). The detector is designed to be placed on the surface of the breast, with the patient in the supine position. Figure 4 shows the positional relationship between the microwave imaging device and the patient.

During imaging, the patient lies supine on the examination bed, with the breast exposed. The device is placed on the skin of the breast under examination and held in place by the operator. The antenna array emits GMP signals to illuminate the breast and receives the reflected signal in turn. Using a step motor, the antenna array rotates from 0° to 360° degrees in 9° steps. In total, 40 sets of 2048 data are acquired, and the total time is 15 min. The received signals are converted from analog to digital via a 12-bit analog-to-digital converter and a confocal image is constructed with the setting conditions of effective permittivities of 6.0 and a sensitivity threshold of 0.7–0.8. The microwave imaging examinations were performed the day before surgery by a radiologist specialized in breast imaging. During examinations, radiologists stood beside the patient and observed for any abnormalities.

### 2.4. Pathological Evaluation

All specimens were sliced into 5- to 10-mm contiguous sections, with thinner slices added by the pathologist as necessary. All diagnoses were made pathologically, and the following histological features were recorded: t histological type, hormone status [estrogen receptor (ER), progesterone receptor (PR), human epidermal growth factor receptor (HER)-2], Ki-67, and tumor size of the total lesion and the invasive component.

### 2.5. Image Analysis

All images were reviewed retrospectively in random order by a radiologist with 11 years of experience in breast imaging, who was told only that the patient had breast cancer. MG, US, and MRI findings and categories were evaluated based on the Breast Imaging Reporting and Data System (BI-RADS) [19]. The presence or absence of FDG uptake was evaluated visually and the maximum standardized uptake value (SUVmax) was measured using a circular region of interest (ROI). The radiologist evaluated whether the microwave imaging was able to detect breast cancer, and compared the location and size of breast cancer lesions on microwave imaging with the pathological findings and those on other imaging modalities, using the 5-point scale (1 = very poor, 2 = poor, 3 = fair, 4 = good, and 5 = excellent). A radiologist evaluated on a 5-point scale

After acquiring the data in two dimensions, the microwave image was image-processed by a computer to map the tumor site in three dimensions.

## 3. Results

Nine patients were included in this study. No adverse events or equipment malfunctions occurred during the investigation. All patients in the study were Japanese.

Table 1 shows the clinical and pathological characteristics of the patients. One patient had bilateral breast cancer (Cases 1, 2) and the remaining eight had unilateral breast cancer (Cases 3–8). Median age was 66.0 years (range, 37–78 years). There were eight cases of invasive ductal carcinoma (IDC) and two cases of ductal carcinoma in situ (DCIS). Neoadjuvant chemotherapy was performed for one case (Case 4) and endocrine therapy for two cases (Cases 1 and 2) before surgery. Median pathological invasive size was 17.5 mm (range, 0–120.0 mm). Median pathological whole tumor size was 60 mm (range, 14.0–120.0 mm). Eight cases were ER positive, seven cases were PR positive, and two cases were HER2 positive. The median value of Ki-67 was 17.5% (range, 3.0–50.0%).

In case 4, neoadjuvant chemotherapy has done and the imaging data is after neoadjuvant chemotherapy before the day of surgery. In case 5, the patient had renal dysfunction and was unable to receive chemotherapy.

Table 2 lists the MG, US, MRI, and PET/CT imaging findings of all patients. Of all 10 cases that underwent MG and US, the lesion could be detected by MG in 8 (80.0%) and by US in 10 (100%). Two cases were undetectable on MG due to dense breast tissue (Cases 8, 9). Breast cancer was detected in all eight cases (100%) that underwent DCE-MRI. Five of the six cases (83.3%) that underwent PET-CT were positive for FDG uptake. A small invasive cancer (infiltration diameter = 0.6 mm) was negative for FDG accumulation (Case 8). Median lesion diameter on US was 28.0 mm (range, 14–70 mm), whereas that on MRI was 35.5 mm (range, 16–60 mm).

Table 3 lists the microwave imaging findings. The microwave imaging system detected breast cancer in all nine cases (100%). The median score of concordance between microwave imaging and breast cancer lesions was 4.5 (range, 4–5) or lesion location and 4.0 (range, 3–5) for lesion size. From the images created, the size of the tumor on the device was described, albeit in approximate size in Table 4. Median lesion diameter on microwave images was 20 mm (range, 5–30 mm). The maximum diameter was measured for those drawn in multiples, and the maximum diameter of the aggregated ones was measured for those in which small ones existed together. Figure 5 shows the positional relationship between the device and the breast. Representative cases are shown in Figure 5 (Case 6) and Figure 6 (Case 8).

No adverse events were reported by the patient during and after the study.

## 4. Discussion

This study examined the ability of a compact microwave radar-based breast device to detect breast cancer in real patients. Images could be obtained in all patients and no adverse events or device malfunctions occurred during the examination period. No adverse events were reported by the patient during and after the study. Among the imaging modalities, breast cancers were detected in 10/10 (100%) by microwave imaging, 8/10 (80.0%) by MG, 10/10 (100%) by US, 8/8 (100%) by MRI, and 5/6 (83.3%) by PET/CT.

The results reveal three important advantages of the microwave imaging device. First, it is useful for detecting breast cancer in women with dense breast tissue. MG was unable to detect breast cancer in two cases (Cases 8 and 9) for this reason; however, these tumors could be detected by microwave imaging as well as by US and MRI. This finding indicates the suitability of the microwave device for detecting breast cancer in women with dense breasts.

Second, the device can help in the detection of micro-invasive ductal carcinoma and DCIS. Case 8 was a predominantly intraductal lesion of diameter 60 mm with invasive diameter of 0.6 mm. Cases 3 and 9 were DCIS with tumor sizes of 45 and 60 mm, respectively. In Cases 3, 8, and 9, MG and PET/CT detected breast cancer in only one-third and in one-half of the cases, respectively. Microwave imaging was able to detect breast cancer in all of these three cases. It is known that micro-invasive cancers (predominantly intraductal lesions or DCIS) are difficult to detect with MG if calcification is not present [25]. These tumors are also known to have low glucose metabolism and are difficult to detect by PET/CT [9,10]. The microwave imaging device may have superior ability to detect these tumors compared with other modalities.

Third, the device appears to be safe, as all examinations were completed with no patient experiencing an adverse event. Furthermore, because microwave imaging does not produce ionizing radiation, unlike MG and PET/CT, patients do not have to worry about radiation exposure. In addition, as the device does not require the use of contrast agents, there is no need to be concerned about the deposition of gadolinium products in the body, which is a concern in contrast-enhanced MRI. We consider that microwave imaging would be suitable for use in repeat annual breast screening.

Several previous studies have attempted to detect breast lesions using microwave imaging. Preece et al. used a microwave imaging system in 86 patients (20 with cancer, 25 with cysts, and 20 with other breast pathology). In their study, the microwave components and supporting mechanical parts were fully integrated in a bed cabinet design. The position of the antenna array was adjustable with the patient positioned on the bed. Patients underwent this examination prior to biopsy or surgery, and the device detected breast lesions in 64/86 (74%). Of these 86 patients, MG was available for comparison in 66. Breast lesions were detected in 49/66 (74%) by microwave imaging and in 51/66 (77%) by mammography (similar detection performance) [26]. A disadvantage of their method is the large footprint occupied by the equipment, which makes it unsuitable for portable use.

Table 4 summarizes the comparison of the experiments with the past. Islam et al. have developed a portable breast cancer detector that uses a similar principle. Although not used in real humans, breast phantoms were able to detect breast cancer. However, the time required for the test was 50 min, which was longer than our study [27].

Luc et al. so explain a device called Wevlia that uses microwaves. The principle is the same as this device, but it differs from this study in that it is a stationary device. The subsystem consists of a 3D stereoscopic camera placed below the table, which is scanned to reconstruct the external surface of the breast and calculate breast volume, with the patient lying in prone position. The MBI scan is then performed using low-power, non-ionizing microwaves that propagate through the breast [28]. There are some reports of actual use of Wevlia on humans.

Cynthia et al. used Wavelia for 24 females using a similar principle. Six were invasive ductal carcinoma, five were invasive lobular carcinoma (ILC), four had benign lesions, and eight patients had a breast cyst. Of the six IDCs, four were detected correctly. Among the ILC lesions, Wavelia correctly localized seven of the nine detected cancerous lesions. The two IDCs that were not detected were less than 10 mm in size. Of the 13 benign cases, 12 were correctly detected lesions and it accurately approximated the location for 10 of these. A total of 92% of patients say they recommend this test method. The inspection time is as long as 50 min [20,21,22]. A Quadratic Discriminant Analysis (QDA) classifier was trained in a 3D feature space to discriminate malignant from benign lesions. Two radiologists reviewed the results. The QDA accurately separated benign from malignant breast lesions in 88.5% of cases. The addition of MBI and the Wevlia malignancy risk calculation was deemed useful by the two radiologists in 70.6% of cases. The sensitivity of the cancer was not 100%. It was also different in that the device was a stationary type.

Lorenzo Sani et al. conducted an experiment using a stationary breast cancer detector called The Mammo Wave system, which uses a similar microwave, for humans. This device was used for 51 breasts with abnormalities on imaging examination and 50 breasts with no findings. Of the 51 cases, 17 were malignant and 30 were benign. The sensitivity of lesion detection using the device was 74%, and the sensitivity of malignant lesion was 71%. In the results of 22 cases of high-density breasts, the sensitivity of lesion detection was 82% and the sensitivity of malignant lesions was 85% [23]. The equipment used in this study was also stationary and different from this study. In addition, although we were investigating including benign lesions, the sensitivity of malignancy was not 100%.

Aleksandra et al. used the device of SAFE (Scan and Find Early). SAFE is a novel microwave imaging device intended for breast cancer screening and early detection. One-hundred fifty patients were enrolled in this study. Sixty-six were benign, eight were high-risk, and eight were malignant. They underwent the SAFE scanning procedure, and the resultant images were compared with other imaging tests such as MRI, mammography, and echo in order to determine the correct detection rate. A sensitivity of 63% was achieved. The sensitivity in malignant lesions was 59%. Their device was also stationary and the sensitivity to malignancy was not 100% [24].

None of the previous studies showed 100% sensitivity to malignant depiction, but in this study, all the lesions were visible. This may be related to the large lesions and the small breasts of Japanese females.

Sasada et al. conducted a survey of five patients with histologically confirmed breast cancer of diameter <1 cm using the same microwave imaging device as ours, but at different facilities. All five of the targeted breast tumors were detected using the device, despite one patient having dense breast tissue, and micro-invasive ductal carcinoma was detected in another case [18]. In agreement with the findings of Sanada et al., the device also correctly detected tumors in the larger number of patients in the present study. This study was conducted at another facility with a larger number of people and proved to be effective in detecting cancer.

In our earlier versions of the microwave imaging device, the dimensions of Prototype-1 were 450 × 300 × 121 mm [16] and those of Prototype-2 were 285 × 225 × 95 mm [15]. Following development of the essential modules in microwave imaging (pulse generator, switching matrix and sampling module using CMOS integrated circuits), the size of the device reported here is 191 × 177 × 188 mm. Although we have been able to greatly reduce the size of the device, our aim in the future is to reduce it further, to the size of a smartphone, which would enable it to be easily used at home. Advances in technology may lead to the situation where breast screening does not need to be conducted only at the hospital.

Although it is a small number of cases, detailed contrasts were made with pathology and multiple modality. We were able to discuss the features and usefulness of this device. There are some limitations in our study. First, we did not evaluate the imaging ability of the device for normal breast or benign tumors. In the present patients, there were no benign breast lesions, only cancer in preoperative imaging and postoperative pathology. It is necessary to evaluate the diagnostic performance of the device in normal breasts and in benign tumors. As a screening device, it is important to distinguish between malignant tumors, benign tumors, and normal breast tissue. This study was conducted prospectively and had a small number of cases. Large-scale prospective studies are needed to identify the most effective strategies for using this microwave imaging system in clinical practice. Normally, we should do a statistical study, but this time due to the small sample size, no statistical studies were conducted.

In the future, if the number of samples increases, we would like to conduct a statistical study.

Second, it is necessary to further develop the image reconstruction methods. The confocal image depends on the setting conditions of effective permittivities of 6.0 and a sensitivity threshold of 0.7–0.8. Although the present score of concordance between microwave imaging and breast cancer lesions for location (median; 4.5, range, 4; good to 5; excellent) was high, that for size (median; 4.0, range, 3; fair to 5; excellent) was relatively low, possibly due to the image reconstruction conditions. Breast size and density, and lesion location and size differ among patients. To detect lesion size and extent of invasion in more detail, it is necessary to optimize settings for each patient in image reconstruction. The image was evaluated by a single radiologist, which may be subjective.

Finally, the examination time is very long (15 min per examination for data acquisition and several hours with a laptop computer for image processing). To examine a large number of patients and improve cost-effectiveness, it is necessary to greatly shorten the data collection time. In addition, the reconstructed images should be checked immediately after the examination, as body movement or mechanical malfunctions during the examination may prevent the collection of optimal image data.

## 5. Conclusions

The microwave imaging device assessed in the present study could detect all 10 breast cancers evaluated, including DICS and micro-invasive cancer. The device was safe and thus appears suitable for repeated use. These findings suggest that the device may be suitable for breast cancer screening.

## Figures and Tables

**Figure 1 diagnostics-12-00027-f001:**
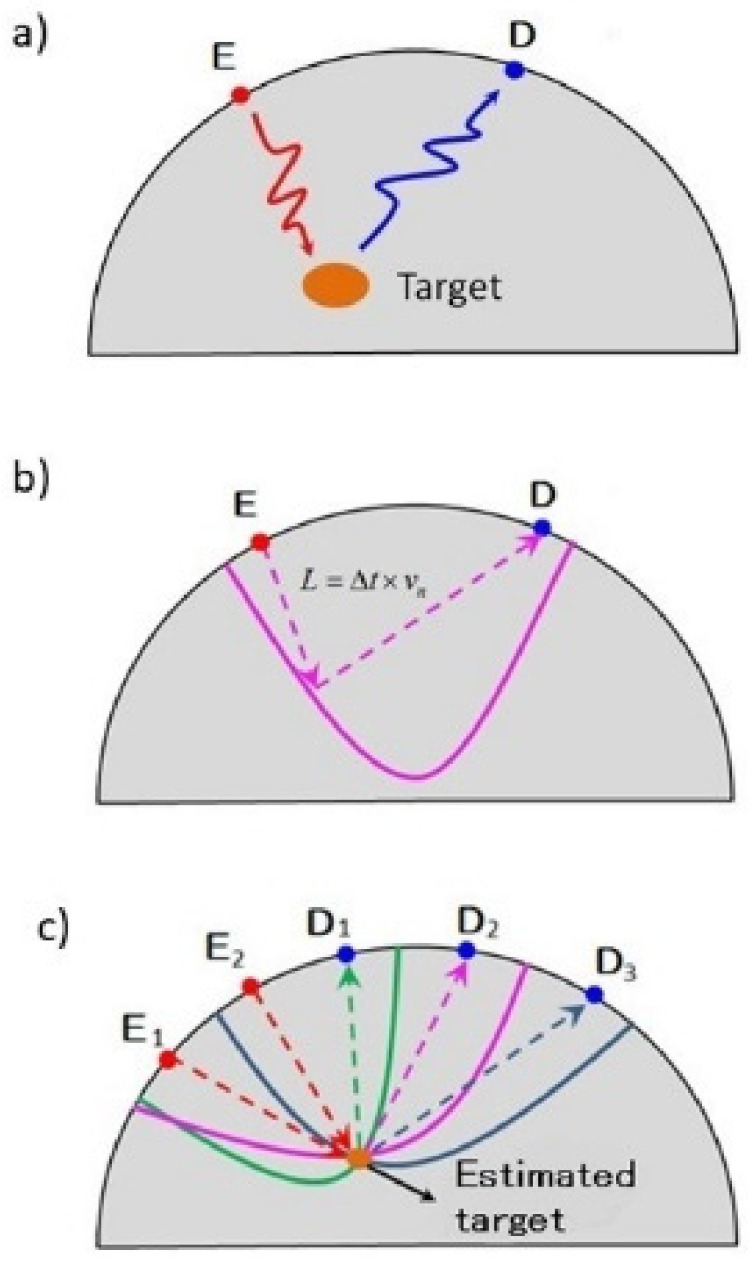
Principle of microwave radar-based breast imaging technology. Microwave radar-based breast imaging technology utilizes the principle of radar, in which radio waves reflect at the interface between target and normal tissues using transmit antenna D and receive antenna E, which have different permittivities (**a**). By calculating the time of flight between the transmitting and receiving antennas, we can infer a target on an elliptical trajectory focusing on these two antennas (**b**). The position of the target can be estimated by computing the intersection of the trajectories of multiple antennas (**c**).

**Figure 2 diagnostics-12-00027-f002:**
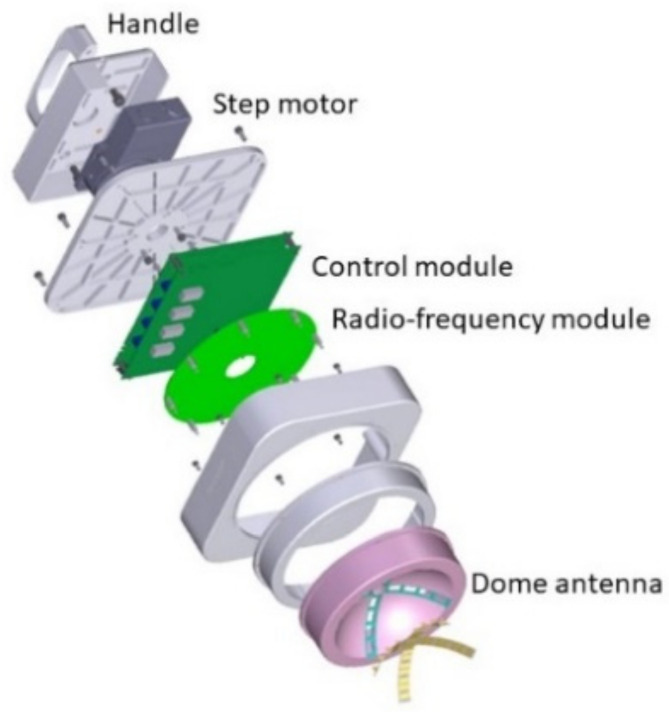
Components of the handheld microwave imaging device. The microwave imaging device comprises a handle, stepping motor, control module, radio-frequency module, and dome antenna.

**Figure 3 diagnostics-12-00027-f003:**
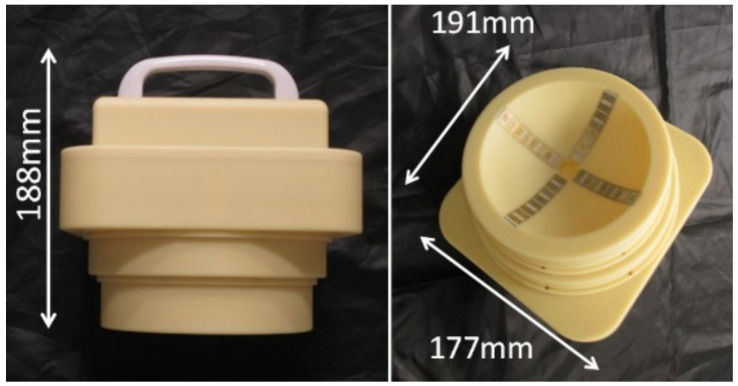
Appearance of the handheld microwave imaging device. The device weighs 2 kg and has dimensions of 191 × 177 × 188 mm.

**Figure 4 diagnostics-12-00027-f004:**
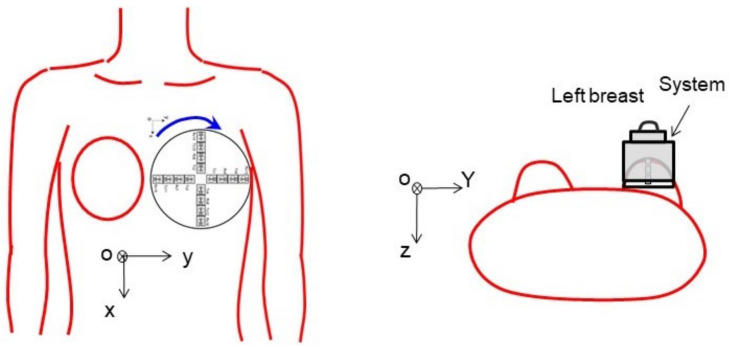
Positional relationship between the microwave imaging device and the patient. The detector is designed to be placed on the breast with the patient in the supine position.

**Figure 5 diagnostics-12-00027-f005:**
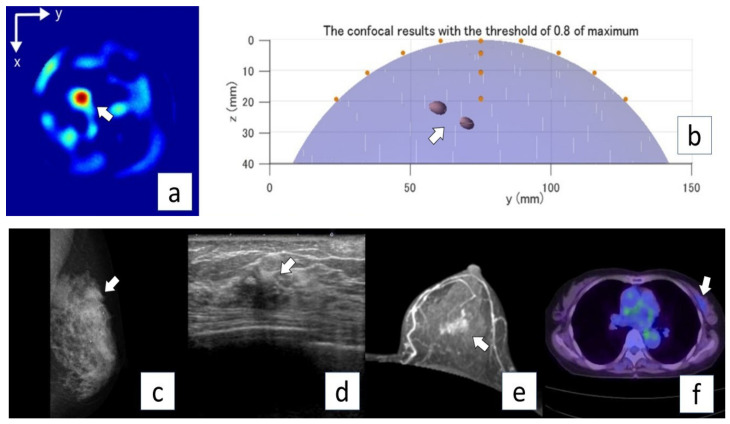
Representative case (Case 6). A 73-year-old woman with invasive ductal carcinoma (invasive lesion, 14.0 mm; total lesion, 40 mm) in the left breast. MG mediolateral oblique view (**a**) shows a focal symmetric density in the upper aspect of the lesion (arrow). US (**b**) and MRI (axial plane, contrast MRI) (**c**) show an irregular mass in the left breast (arrows). PET/CT (**d**) shows mild FDG uptake in the left breast (arrow). Microwave imaging 2D-coronal view (**e**) and 3D-axial view (**f**) also show the breast cancer lesion in the left breast.

**Figure 6 diagnostics-12-00027-f006:**
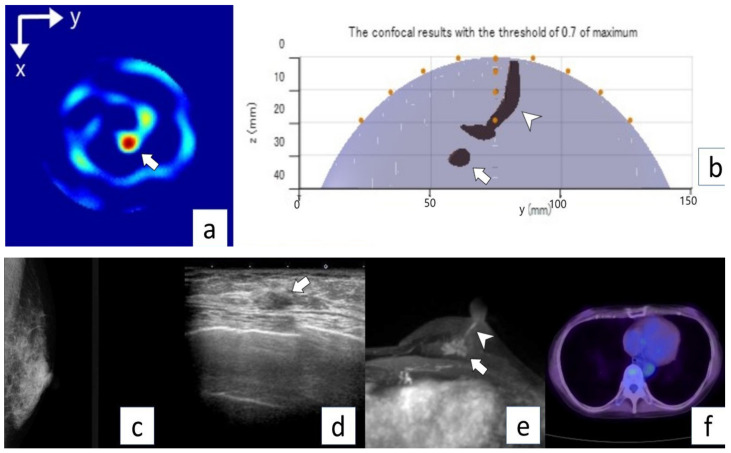
Representative case (Case 8). A 73-year-old woman with invasive ductal carcinoma (invasive lesion, 0.6 mm; total lesion, 60 mm) in the left breast. No abnormal lesion is seen on MG mediolateral oblique view (**a**). US shows an irregular mass in the left breast (arrow) (**b**). MRI (**c**) shows an irregular mass (arrow) and intraductal lesion extending toward the nipple (arrowhead). No abnormal lesion is seen on PET/CT (**d**). Microwave imaging 2D-coronal view (**e**) and 3D-axial view (**f**) also show breast cancer (arrows).

**Table 1 diagnostics-12-00027-t001:** Characteristics and pathological factors of the patients.

No of Case	Age	BMI	Tumor Side	Histology	Tumor Size (mm)		ER	PR	HER2	Ki 67 (%)
					Invasive Lesion	Total				
1	69	25.9	R	IDC	28.0	112.0	+	+	-	4.8
2			L	IDC	38.0	85.0	+	+	-	35.2
3	78	17.2	R	DCIS	0	45.0	+	+	+	6.7
4	37	25.4	R	IDC	60.0	60.0	+	+	-	3.0
5	66	19.1	R	IDC	120.0	120.0	+	+	-	32.7
6	73	19.0	L	IDC	14.0	40.0	+	+		12.7
7	47	27.8	L	IDC	14.0	14.0	+	+	-	22.7
8	60	17.3	L	IDC	0.6	60.0	+	+	-	15.0
9	40	20.6	R	DCIS	0	60.0	-	-	-	20.0
10	67	20.2	L	IDC	21.0	60.0	+	+	+	50.0

**Table 2 diagnostics-12-00027-t002:** Image findings of MG, US, MRI, and PET/CT of the patients.

No of Case	MG			US			MRI			PET/CT
	Breast Density	Finding	Cat	Finding	Size (mm)	Cat	Finding	Size (mm)	Cat	SUVmax
1	B	Mass	5	Mass	16	5	Mass	16	5	NA
2	B	Mass	5	Mass	23	5	Mass	46	5	NA
3	B	FAD	4	Mass	40	5	Mass	33	5	4.4
4	C	Mass	5	Mass	37	5	Mass	42	5	5.4
5	B	Mass	4	Mass	70	5	NA	NA	NA	NA
6	C	FAD	4	NML	19	4	NME	24	4	1.6
7	B	Mass	5	Mass	14	5	NA	NA	NA	5.7
8	C	Negative	1	NML	22	4	NME	28	5	Negative
9	D	Negative	1	NML	38	5	NME	38	5	NA
10	D	Calcification	4	Mass	33	5	Mass	60	5	4.7

MG: Mammography, A: Almost entirely fatty, B; Scattered fibroglandular density, C: Heterogeneously dense, D: Extremely dense, FAD: focal symmetric density, NML: Non-mass lesion, NME: Non mass enhancement, NA: Not applicable.

**Table 3 diagnostics-12-00027-t003:** Characteristics of microwave imaging.

No of Case	Detection	Consistency with Cancer Lesion		Size (mm)
		Location	Size	5
1	positive	5	5	5
2	positive	5	4	15
3	positive	4	5	15
4	positive	4	4	20
5	positive	4	4	30
6	positive	5	4	20
7	positive	5	5	5
8	positive	4	4	30
9	positive	5	4	20
10	positive	4	3	20

**Table 4 diagnostics-12-00027-t004:** Comparison with previous experiments.

	Equipment Features (Portable)	Principle	Machine Size, Weight	Number of Cases Total Number of Cases Malignant, Benign	Grades Sensitivity, Specificity, etc.
This study	A prototype of a portable breast cancer detector using a radar-based imaging system	The core functional part of the detector comprises 65-nm technology CMOS integrated circuits covering the ultrawideband width of 3.1–10.6 GHz, which enables the generation and transmission of Gaussian monocycle pulse (GMP) and single port eight throw switching matrices for controlling the 4 × 4 cross-shaped dome antenna array.	Size: 191 × 177 × 188 mm weight: 2 kg	10 cases (All were malignant).	All cancers were detected.
Sanada et al. [18]	A prototype of a portable breast cancer detector using a radar-based imaging system	The core functional part of the detector comprises 65-nm technology CMOS integrated circuits covering the ultrawideband width of 3.1–10.6 GHz, which enables the generation and transmission of Gaussian monocycle pulse(GMP) and single port eight throw switching matrices for controlling the 4 × 4 cross-shaped dome antenna array	Size: 191 × 177 × 188 mm weight: 2 kg	5 cases (All were malignant).	All cancers were detected.
Aleksandar et al. [20]	SAFE (Scan and Find Early) is a novel microwave imaging device. Patients were required to lie prone on the table with one breast inserted into the coupling medium cup. The device is not portable.	Thirty-six receiving transmitter position points, a total of 1296 measurements. The operating frequency band was between 1.4 GHz and 8 GHz.	The size of the device is not stated, but a special cup is embedded in the bed.	115 cases (benign: 66, high-risk: 8, malignant: 41).	Sensitivity: 63%
Cynthia E Keen. [21,22,23]	“Wavelia” including two subsystems, the optical breast contour detection (OBCD) subsystem and the MBI subsystem. The OBCD subsystem consists of a 3D stereoscopic camera placed. The device is not portable.	Eighteen equally spaced wideband Vivaldi-type antennas. Each probe illuminates the imaging domain in turn, while the remaining antennas receive the electromagnetic scattering at various angles around the circle. The probe array also moves at 5 mm intervals.	The size of the device is not stated, but a special cup is embedded in the bed.	24 cases (11; malignant, 13; benign)	sensitivity of malignant; 81%, sensitivity of benign; 92%
Lorenzo Sani et al. [24]	The MammoWave; the hub is internally covered by microwave absorbers. The hub is equipped with a hole with a cup, allowing the insertion of the patient’s breast, with the patient lying in a prone position. The device is not portable.	Consists of an aluminum cylindrical hub containing two antennas, one transmitting and one receiving antenna, which operate in the 1–9 GHz frequency band.	The size of the device is not stated, but a special cup is embedded in the bed.	103 breasts with no radiological finding (NF) and radiological findings (WF).	a sensitivity of 74% (the sensitivity of lesion detection using the device was 74%, and the sensitivity of malignant lesion was 71%. malignant lesions was 85%).

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
