# Peer review of "Feasibility of Portable Microwave Imaging Device for Breast Cancer Detection"

_diagnostics, 2021, doi:10.3390/diagnostics12010027_

Round 1

Reviewer 1 Report

Title: Feasibility of portable microwave imaging device for breast cancer detection
Diagnostics Manuscript ID: diagnostics-1457405

This study aims to investigate the feasibility and safety of a portable microwave breast imaging device in clinical practice.

This manuscript consists of a structured abstract with keywords, 5 sections (introduction, materials & methods with 4 subsections, results, discussion, and conclusions) on 8 pages of single-spaced text with embedded tables and figures. There are 21 references, 5 figures, and 3 tables. No URL is cited. No appendices or supplements. 

Microwave breast imaging systems are under development at many laboratories worldwide, and this report originates from an important pioneering group of universities and health centers in Japan. Several of their prior reports are cited, but the need for further feasibility testing is uncertain since similar technology has been demonstrated many times in the recent past. No information regarding safety is presented. It should be made clear: What is new? This should be done in the context of many predecessor systems to identify and emphasize the unique aspects and improvements in the reported new system. 

The 10 patients imaged in this study have large size (4-12 cm) breast masses and no normal or benign lesions were studied. The race and ethnicity of the study population are not described. Patient demographics including BMI and outcomes are not tabulated. 

Table 2 combines many modalities (MG, US, MRI, and PET/CT) without showing any significant inter-modality variation. This grossly oversimplifies the subjective image analysis. "Consistency with cancer lesion" reported in the Tables 2 and 3 is a very weak metric for cancer imaging system performance. This subjective metric was defined on lines 164-166 of page 5, but no examples are illustrated. Observer variation is not reported.  Apparently, all observations were made by a single radiologist. No justification was provided for a single observer. 

The sentence on lines 167-168 on page 5 regarding mapping tumor size in 3D is vague. No example is shown. The images are not available (e.g., through URL for example) for independent review. RECIST criteria were not used. Why not? 

There is no comparison of tumor size measured on the pathological specimen and various modalities, including microwave images. 

"We are satisfied with this result in 10 cases." (line 272, page 9)  This statement is inappropriate in a scientific publication. The authors' level of satisfaction is superfluous. 

Only 2 of 21 references are from 2020. Many important recent references were omitted from the bibliography. The following report was not cited and discussed. How does this system differ from the authors' approach?

M. T . Islam, et al. A Low Cost and Portable Microwave Imaging System for Breast Tumor Detection Using UWB Directional Antenna array. Scientific Reports | (2019) 9:15491. 

The following report appears to originate from the same authors as the current report, but the differences in technology and approach are not clear. This report should be cited and the differences in the current report should be clearly explained: 

Shinsuke Sasada, et al. Microwave Breast Imaging Using Rotational Bistatic Impulse Radar for the Detection of Breast Cancer: Protocol for a Prospective Diagnostic Study. JMIR Res Protoc 2020;9(10):e17524) doi: 10.2196/17524 

According to CE Keen (2021), as of 2020, 10 MBI prototypes had been clinically tested.

Ref. Cynthia E Keen. Microwave imaging could provide safer, more comfortable breast cancer screening. https://physicsworld.com/a/microwave-imaging-could-provide-safer-more-comfortable-breast-cancer-screening/#:~:text=Microwave%20breast%20imaging%20%28MBI%29%20represents%20a%20promising%20non-invasive,2020%2C%2010%20MBI%20prototypes%20had%20been%20clinically%20tested. 14 Sep 2021

Several important and relevant recent microwave breast imaging system developments have been published but these reports were not cited and discussed in the current manuscript. Recently, numerous microwave breast imaging system papers have described these systems including the Wavelia, MammoWave, and SAFE systems, among others:

BRIAN M MOLONEY, et al. The Wavelia Microwave Breast Imaging system–tumour discriminating features and their clinical usefulness. Br J Radiol 2021; 94: 20210907. 

Brian M. Moloney, et al. Microwave Imaging in Breast Cancer – Results from the First-In-Human Clinical Investigation of the Wavelia System. Acad Radiol 2021;&:1–12. 

Luc Duchesne, et al. Wavelia Microwave Breast Imaging: Identification and Mitigation of possible Sources of Measurement Uncertainty. 2019 - 13th European Conference on Antennas and Propagation IEEE (EuCAP).

Lorenzo Sani, et al. Breast lesion detection through MammoWave device: Empirical detection capability assessment of microwave images’ parameters. PLoS ONE 16(4): e0250005 (2021) 

Aleksandar Janjic, et al. SAFE: A Novel Microwave Imaging System Design for Breast Cancer Screening and Early Detection —Clinical Evaluation. Diagnostics 2021, 11, 533.

Prior work on microwave imaging should be mentioned and the technology used in the current report should be compared to emphasize the important differences and to explain what, if any, advances were made. 

The sample size is small (n-10). There is no formal statistical experimental design and analysis of the results. 

Overall, this manuscript introduces a comparison of microwave breast imaging obtained with a "portable" probe to pathological mammectomy specimens of advanced cancers and DCIS. Subjective assessments of imaging results obtained with several modalities were reported (MG, US, MRI, and PET/CT) revealing insignificant differences for relatively large tumors. The metrics used in this study, both subjective and objective, were relatively weak showing minimal variations. Several other microwave breast imaging systems have been introduced in recent years, but there is no comparison with the new technology in this report. Feasibility of microwave imaging to detect advanced large cancers and DCIS is an issue of dubious importance. Many experimental details require clarification. The bibliography should be updated. 

Author Response

We thank the Reviewer for this kind and relevant comment. We have revised the manuscript based on your helpful comments.

We added the information that all the patients were Japanese and BMI to Table 1.

Statistics are omitted due to the small number of studies.

We apologize that table 2 has disappeared, so we added it. The size of the tumor in each image is shown.MG, US, MRI, and PET/CT. All the image inspections performed are described.

The equipment is in the prototype stage in the first place. Since this test is conducted on cases with a diagnosis of cancer, we have only examined whether it is possible to visualize it. So it includes cancers of various properties such as invasive cancer, non-invasive cancer, and various sizes of cancer.

The image is interpreted based on BI-RADS and is described in the text.

As your say we erased the sentence, "We are satisfied with this result in 10 cases."

This device did not measure the size of the tumor.

We would like to make it an issue for future study.

Thank you for your lot of references.

It differs from previous studies in that the only reports that cancer could be detected in all cases were sloppy reports.

In addition, the number of inspections has increased compared to the pervious report by Sanada. We added to those to the text.

Normally, as you pointed out, statistical analysis should be performed.

Since the number of samples was small and statistical analysis was not possible, we did not perform statistics this time. We that as a limitation to the text.

Reviewer 2 Report

In this manuscript, the authors report the application of radar technology for the screening of breast cancer. A fully functional device was used and the results were compared with those of standard medical analysis. Ten breast cancers were analyzed with positive results.

The article is well written. The topic is interesting and the results are very promising.

Detailed comments

There are some problems with the figures and the captions. See Figures 1, 5, and 6.

Also in Figures 5 and 6, the lettering is wrong. I have highlighted the parts in the attached file.

Author Response

We thank the Reviewer for this kind and relevant comment. We have revised the manuscript based on your helpful comments.

I changed the lettering of figure5 and 6.

The explanation of the transmitting antenna and the receiving antenna was added to the explanation of Fig1.

Reviewer 3 Report

I have only a few questions and corrections:

  • Authors state that all patients referred for mastectomy between sept 2017 and march 2018 were included in the study. Yet, they had only 9 patients. Why was the study group so small? These were all the patients referred for surgery? Or were there also other exclusion criteria than the ones mentioned (pregnancy, breastfeeding etc)?
  • Because of the large size of the lesions (measured how? On mammography, ultrasound, or pathology?) I think it is not appropriate to say that microwave is suitable to detect breast cancer in screening patients as long as on mammography (+/- ultrasound) screening we detect lesions smaller than 1 cm. If the size was measured on pathology specimens, I suggest giving also the pathology and the mammography and ultrasound sizes (also because sizes as 60 or 120 mm on imaging methods would contraindicate the surgery as the first therapeutic step – these patients should have undergone neoadjuvant chemotherapy…).
  • And yes, because micro-wave was also able to identify DCIS – this is indeed promising but there were only 2 cases. We cannot draw any conclusion
  • In table 1, at case 8, the size of the invasive component is 0.6 mm, while in the discussions the size is 6 mm (row 241)
  • Figures 5 and 6 – the notation of the figures does not match the legend. Microwave images are noted a and b, in the legend a and b are the mammogram and the ultrasound. Should be corrected. And before figure 6 appears “(f) also show the breast cancer lesion in the left breast. “ But the PET-CT was already mentioned.

Author Response

We thank the Reviewer for this kind and relevant comment. We have revised the manuscript based on your helpful comments.

Since we did not know if this device would work properly, we first conducted a pilot study in 10 cases. There was no other exclusion other than those mentioned in the paper.

There were more patients who underwent surgery in our institution.

We will plan to increase the number of cases in the future.

We measured lesions by mammography, ultrasound and pathology.

We measured tumor by ultrasound, MRI and pathology. Image information was added to Table 2. We apologize for the disappearance of Table 2. I added Table 2. At first we did not know the device would work correctly, So we designed the study for patients who would undergo surgery of mastectomy. As you say, it cannot be called screening. This time, we wanted to know if this device could detect breast cancer normally, so we used this device regardless of its size.

In Case 4, neoadjuvant chemotherapy has done, so we added the information. In case5, this size should have been done neoadjuvant chemotherapy, but the patient had renal dysfunction and was unable to receive chemotherapy. We added that information, too.

We apologize that the size of case8 was wrong, we changed the size from 6mm to 0.6mm in row241.

We changed Fig 5 and 6 of the notation and figures. Microwave images are noted e and f following notation.

Round 2

Reviewer 1 Report

The authors respond to the initial critiques with a general (not itemized) list of comments and changes. The authors' response to the review is exceedingly brief relative to the length and detail in the review. Failure to address the review point-by-point with sufficient detail to justify the authors' response is very disappointing. 

The changes made in the manuscript corresponding to each point in the critique are not highlighted. Why not? 

Some issues were dismissed and not addressed: "Statistics are omitted due to the small number of studies.", and "This device did not measure the size of the tumor.".  One response mentions "sloppy reports.". What does this mean?

The race and ethnicity of the study population are not explained in the abstract.  The range of tumor sizes in the study population is not described in the abstract. 

The authors do not acknowledge that all 10 cases they included are advanced stage breast cancer (tumors from 4-10 cm). No early stage disease was considered. This is a MAJOR limitation, but OK for feasibility testing.

However, the size of the tumor in Case 8 should be 0.6 mm. The various parts of Figure 6 which pertain to Case 8 do not have scale bars. The orientation of figure 6a relative to other parts of the figure is difficult to understand since no other show an x-y axis at a corresponding location. 

The justification for feasibility testing of microwave breast cancer imaging is questionable since this technique has been described many times in the past.

As mentioned in the review, many authors have published prior work on this topic, but there is no tabulation of comparative systems and a detailed list of specifications to distinguish the authors' new instrument from prior work. This is a MAJOR limitation. 

In line 270, "In agreement with the findings of Sasada et al," is mentioned, but no reference is cited. I presume that this should be [18]. Given that the same instrument was used, there is no hint of what is new in the current report. 

The comment that "We are satisfied with this result in 10 cases." on line 272 is inappropriate and should be removed. This can be restated in a more appropriate form. 

"The aim of this study was to assess the ability of a newly developed device that uses non-invasive radar-based microwave technology to detect breast cancer in real patients."  appears in lines 71-73. Wasn't this already demonstrated in reference [18]? 

Author Response

We wish to express our appreciation to the Reviewer for the insightful comments, which have significantly helped us to improve the paper.

The changes made in the manuscript corresponding to each point in the critique are not highlighted. Why not? 

→We were sorry, it's hard to understand. Last time, I fixed it with red, but I also highlighted it with yellow.

Some issues were dismissed and not addressed: "Statistics are omitted due to the small number of studies.", and "This device did not measure the size of the tumor.".  One response mentions "sloppy reports.". What does this mean?

→This time, we conducted the research under the submission to the Institutional Review Board, which is to conduct 10 studies, so we cannot increase it any more. It is described in the text as a limitation.

The approximate size of the tumor was measured from the completed image and is shown in Table 3.

The race and ethnicity of the study population are not explained in the abstract.  The range of tumor sizes in the study population is not described in the abstract. 

→The size the tumor and race of are listed in the abstract.

The authors do not acknowledge that all 10 cases they included are advanced stage breast cancer (tumors from 4-10 cm). No early stage disease was considered. This is a MAJOR limitation, but OK for feasibility testing.

However, the size of the tumor in Case 8 should be 0.6 mm. The various parts of Figure 6 which pertain to Case 8 do not have scale bars. The orientation of figure 6a relative to other parts of the figure is difficult to understand since no other show an x-y axis at a corresponding location. 

→The size of invasive diameter is 0.6 mm, but the cancer is 60 mm including intraductal lesions.

It is described in the table in the text. The text is highlighted in blue. Please confirm.

The justification for feasibility testing of microwave breast cancer imaging is questionable since this technique has been described many times in the past.

As mentioned in the review, many authors have published prior work on this topic, but there is no tabulation of comparative systems and a detailed list of specifications to distinguish the authors' new instrument from prior work. This is a MAJOR limitation. 

We have included a table of comparisons with past studies in Table 4.

Of the documents you taught, the documents 24-26 in the text were the same experiment, so I listed them together.

In line 270, "In agreement with the findings of Sasada et al," is mentioned, but no reference is cited. I presume that this should be [18]. Given that the same instrument was used, there is no hint of what is new in the current report. 

→We added “In agreement with the findings of Sanada et al, the device also correctly detected tumors in the larger number of patients in the present study. This study was conducted at another facility with a larger number of people and proved to be effective in detecting cancer.”

The comment that "We are satisfied with this result in 10 cases." on line 272 is inappropriate and should be removed. This can be restated in a more appropriate form. 

→We deleted "We are satisfied with this result in 10 cases."  and added Although it is a small number of cases, detailed contrasts were made with pathology and multiple modality. We were able to discuss the features and usefulness of this device.

"The aim of this study was to assess the ability of a newly developed device that uses non-invasive radar-based microwave technology to detect breast cancer in real patients."  appears in lines 71-73. Wasn't this already demonstrated in reference [18]? 

→We added" in more cases, including cancers of various sizes and pathologies, and whether they can be used in a different facility "  after the sentence,  " The aim of this study was to assess the ability of a newly developed device that uses non-invasive radar-based microwave technology to detect breast cancer in real patients." 
